# Effects of Cesium on Physiological Traits of the Catherine’s Moss *Atrichum undulatum* Hedw.

**DOI:** 10.3390/plants13010054

**Published:** 2023-12-23

**Authors:** Jelena N. Stanojković, Marija V. Ćosić, Djordje P. Božović, Aneta D. Sabovljević, Marko S. Sabovljević, Ana A. Čučulović, Milorad M. Vujičić

**Affiliations:** 1Institute for the Application of Nuclear Energy—INEP, University of Belgrade, Banatska 31b, 11080 Zemun, Serbia; anas@inep.co.rs; 2Faculty of Biology, Institute of Botany and Botanical Garden Jevremovac, University of Belgrade, Takovska 43, 11000 Belgrade, Serbia; marijac@bio.bg.ac.rs (M.V.Ć.); djordje.bozovic@bio.bg.ac.rs (D.P.B.); aneta@bio.bg.ac.rs (A.D.S.); marko@bio.bg.ac.rs (M.S.S.); milorad@bio.bg.ac.rs (M.M.V.); 3Faculty of Science, Department of Botany, Institute of Biology and Ecology, Pavol Jozef Šafárik University in Košice, Mánesova 23, 040 01 Košice, Slovakia

**Keywords:** antioxidative enzymes, bryophytes, catalase, oxidative stress, peroxidase, reactive oxygen species, superoxide dismutase

## Abstract

Mosses are proven bioindicators of living environments. It is known that mosses accumulate pollutants from precipitates and, to some lesser extent, from the substrate. In this study, the effects of cesium (Cs) on the physiological traits of acrocarp polytrichaceous Catherine’s moss (*Atrichum undulatum* Hedw.) were studied under controlled, in vitro conditions. Cesium can be found in the environment in a stable form (^133^Cs) and as a radioactive isotope (^134^Cs and ^137^Cs). Belonging to the same group of elements, Cs and potassium (K) share various similarities, due to which Cs can interfere with this essential element and thus possibly alter the plant’s metabolism. Results have shown that Cs affects the measured physiological characteristics of *A. undulatum*, although the changes to antioxidative enzyme activities were not drastic following Cs treatments. Therefore, the activities of antioxidative enzymes at lower pH values are more the consequence of pH effects on enzymatic conformation than simply the harmful effects of Cs. Moreover, Cs did not affect the survival of plants grown on the solid substrate nor plants grown in conditions of light and heavy rain simulation using Cs with variable pH, indicating that Cs is not harmful in this form for the studied species *A. undulatum*.

## 1. Introduction

In the late 1960s, Rühling and Tyler [1] developed the idea of using bryophytes to estimate atmospheric heavy metal deposition. Later on, mosses were employed to map the deposition of ^137^Cs from the radioactive cloud in polluted areas after the Chernobyl accident in 1986, which greatly increased interest in bryophytes as bioindicators of radioactivity. The devastating effect of the accident was a small turning point in the scientific sense, as a series of experimental research studies regarding the influence of stable and radioactive cesium on living organisms were conducted, particularly in plants. However, there is a lack of knowledge on the effects of cesium (Cs) on bryophytes’ morphology and physiology.

Bryophytes represent the second largest group among land plants (after flowering plants) and are divided into three main lineages, namely hornworts (Anthocerotopsida), liverworts (Marchantiopsida), and mosses (Bryopsida) [2]. Due to their specific characteristics, such as the lack of a protective cuticle and the presence of cell walls rich in negatively charged groups, as well as their widespread distribution, several species of bryophytes have been identified as preferable bioindicators [3,4,5,6,7]. Bryophytes display a high tolerance and/or resistance to many substances found in their habitats which can be highly toxic to other plants. Thus, these organisms are commonly used for monitoring changes in environmental conditions, enabling the implementation of suitable measures of prevention, protection, or improvement of environmental quality. Since bryophytes absorb nutrients with their whole surface, their growth depends mainly on atmospheric deposition and rainfall, i.e., the presence of water. Moreover, this is also the way that potentially toxic elements can enter the bryophytes and interfere with the metabolism. In rainy periods when substrate elements dissolve and a thin water film is established along the plantlets, the substrate contributes to the mineral content and element absorption. On the other hand, during dry periods, most species have little or no ability to take up water and other elements from the substrate. While some species can withstand extended periods of air drying without suffering any harm [8], others cannot survive extreme tissue dehydration. High cation exchange capacity (CEC), the absence of cuticle in gametophytes, and rather simple tissue organization enable bryophytes to easily adsorb available elements, including various metal ions [3]. In general, the adsorption of cations depends on their concentration, binding affinity to moss cell walls, duration of exposure to ions, and the pH value of the surroundings [9]. However, the capacity to exchange cations with the environment is the species-specific feature of mosses and greatly depends on their cell wall composition [10]. After adsorption on moss cell walls, metal ions can be transported into the cells via membrane transporters present for other essential elements. Thus, metal ion accumulation can disturb the transport of essential ions and consequently lead to changes in physiological processes. Numerous species are suggested to have the ability to accumulate trace elements, with pleurocarp mosses being preferred since a bigger surface is in contact with substrates. Some acrocarp mosses, such as Catherine’s moss (*Atrichum undulatum* Hedw.) have also been found to accumulate trace metals from the environment [11,12,13,14]. However, there is little information on *A. undulatum*’s capacity for bioaccumulation. Therefore, this robust, widespread, polytrichaceous species that inhabits subneutral soils was selected as the model in experiments with cesium presented in this manuscript. *Atrichum undulatum* has quite big leaves (up to 10 mm) with lamellas present on the costa that can significantly increase the leaf surface and, thus, the potential metal uptake in wet conditions. Those characteristics also imply the high potential of *A. undulatum* to interact with cesium contaminants during rainy periods. 

The mineral pollucite (zeolite) is the main source of stable cesium. The concentration of naturally occurring stable cesium (^133^Cs) ranges from 0.003 to 0.6 μg L^−1^ in water and from <1000 to 30,000 μg kg^−1^ in soils. Substantially higher concentrations (up to 7 g kg^−1^) are found in industrial wastewater and in incineration fly ashes from coal-burning power plants [15]. Radioactive isotopes of cesium (^137^Cs and ^134^Cs) are formed in nuclear fission reactions mostly from different nuclear programs (nuclear power plants, nuclear weapon production and testing), as well as uranium mining and milling and commercial fuel reprocessing [16]. Nuclear accidents, such as those that occurred in Chernobyl and Fukushima, also contributed significantly to the rise of ^137^Cs and ^134^Cs in the environment [17]. The toxic effects of radioactive cesium originate from two sources: the toxic effect of the stable element per se as well as the toxicity coming from the radiation. The unstable radiocesium decays, releasing high-energy ionizing radiation in the form of beta particles and gamma rays that may cause damage (both direct and indirect) in plants. Belonging to the same group of elements, cesium and potassium share various similarities, due to which Cs can compete with this essential element. It disturbs potassium homeostasis, thus leading to potassium deficiency [18]. In addition, Cs mobility in biological systems is relatively high [19]. The effects of Cs exposure on growth, metabolism, and genetics have been reported only for some vascular plants and algae [18,19,20,21,22,23]. Several reports show that high concentrations of Cs damage the chloroplast function due to decreased chlorophyll content and disorder of the fluorescence kinetic parameters as well as inhibition of photosynthesis [24,25,26,27,28]. While low doses of Cs do not seriously damage plant growth, high concentrations have several different impacts on plants: they reduce plant growth, change the ultrastructure of chloroplasts, disrupt photosynthetic pigments, induce accumulation of reactive oxygen species (ROS), and inhibit antioxidant enzymes, consequently leading to oxidative exposure and peroxidation damage to cell membranes [29]. However, no information regarding the impact of Cs exposure, duration period, its accumulation, and stress response on bryophytes can be found. 

In this paper, the effect of cesium on physiological traits in moss *A. undulatum* was studied. The aim of the study was to investigate whether cesium present in the medium and water solution (rain simulation) induces stress in *A. undulatum*. Furthermore, we tested whether the solution pH affects physiological traits of the moss *A. undulatum*.

## 2. Results

### 2.1. Effects of Long-Term Exposure to Cesium Acetate in Experiment I (Uptake from the Solid Substrate)

Long-term exposure of *A. undulatum* to different concentrations of cesium acetate affected catalase (CAT), peroxidase (POX), and superoxide dismutase (SOD) activities in different manners (Figure 1). The CAT activity increased with enlargement of the cesium concentration in the medium, suggesting possible Cs uptake from the medium. The highest increase in CAT activity was detected in plants grown in the media supplemented with 1 mM cesium acetate, which was higher by 110.4% than in control plants (Figure 1C). At the same concentration of cesium in media (1 mM), POX activity was significantly reduced compared to control plants, i.e., POX activity decreased by 18.3% compared to the control plants group (Figure 1B). Similar to POX activity, SOD activity decreased upon Cs treatment, with the lowest activity of 1.97 U mg^−1^ proteins at 1 mM cesium acetate concentration. In comparison to the control plants group, the SOD activity was reduced by 54.8% (Figure 1A).

Quantification of H_2_O_2_ indicated the highest content in plants grown in the medium supplemented with 1 mM cesium acetate, which was 13.5% higher than in control plants (Figure 1E). Lipid peroxidation measured via MDA concentration followed a similar trend as H_2_O_2_. An increase of 29.6% regarding MDA concentration was observed in plants treated with 1 mM cesium acetate compared to values in control plants (Figure 1D). 

The total phenolic content (TPC) decreased by 32.1% and 68.5% in plants grown in media supplemented with 0.5 and 1 mM cesium acetate, respectively. However, when 1.5 mM cesium acetate was applied, the TPC content increased to values similar to those obtained in control samples (7.3 mmol GAE g^−1^) (Figure 1F).

In general, the obtained results indicate that plants uptake Cs from the solid medium and that Cs affected the investigated physiological parameters after five weeks, since there are significant differences between the plants treated with 1 and 1.5 mM cesium acetate and the control group. Moreover, 1 mM of cesium acetate led to the greatest changes in the enzyme activities as well as in the TPC, suggesting that plants engage antioxidative mechanisms to reduce the oxidative stress that potentially arose from the Cs presence in the solid media.

### 2.2. Effect of Short-Term Exposure for 2 h to Cesium Acetate in Experiment II (Light Rain Simulation)

The highest superoxide dismutase (SOD) activity was documented in the control group, i.e., plants kept for 2 h in distilled water at pH 8, while for the i-th addition of cesium acetate, the SOD activity decreased (Figure 2A). Compared to control plants, SOD was decreased by 83.3%, 83.4%, and 67.4% in plants exposed to 0.5, 1, and 1.5 mM cesium acetate, respectively. Besides the significant changes in SOD activity at pH 8, the SOD activity was also noticeably altered in plants kept in water solution supplemented with cesium acetate at pH 6. In that case, the SOD activity was higher in the plants treated with 0.5, 1, and 1.5 mM cesium acetate by 310%, 212%, and 343% respectively, compared to the control plant group. Meanwhile, at pH 3, the SOD activity increased significantly (for 178%) in plants when treated with 1.5 mM cesium acetate compared to control plants. Similar to changes in SOD activity observed in plants at pH 3, the highest cesium acetate concentration at pH 4, significantly increased the SOD activity. In comparison to control plants, the SOD increase was 65.3% at this point (Figure 2A).

Similar to the results obtained for SOD, the highest activity of peroxidases (POX) was observed in the control plants at pH 8, suggesting that the high pH value affected the enzymatic activity unrelated to cesium presence (Figure 2B). When cesium acetate was added, combined with pH 8, the POX activity decreased significantly. Compared to control plants, POX was decreased by 30.2%, 47.3%, and 21.6% in plants exposed to 0.5, 1, and 1.5 mM cesium acetate, respectively. On the contrary, the presence of cesium acetate at pH 6 led to the increase in POX activity compared to the control plants. POX activity was increased by 24.9% in plants exposed to 0.5 mM cesium acetate, 25.2% in plants exposed to 1 mM cesium acetate, and 21.8% in plants exposed to 1.5 mM cesium acetate. Lower pH values (3 and 4) in combination with cesium acetate slightly affected the activity of peroxidase. Therefore, no clear trend in the POX activity was documented (Figure 2B).

In all experimental groups kept for 2 h in water solution with different cesium acetate concentrations and pH values, the highest CAT was observed at pH 6 (Figure 2C). In this case, the CAT activity was increased by 53.4% in plants exposed to 0.5 mM cesium acetate for 2 h, and 31% in plants exposed to 1 mM cesium acetate, compared to control plants. The CAT activity in *A. undulatum* was substantially lower at other tested pH values in combination with different cesium concentrations and was particularly similar in absolute values, suggesting that pH 6 is optimal for catalase activity.

In general, plants exposed to 2 h of light rain simulation showed altered enzymatic activities. Activities of SOD and POX seem to show a similar trend with cesium acetate addition, especially at pH 6 and pH 8. On the other hand, catalases were active only at pH 6, suggesting specific conditions for their activation. Nevertheless, these results indicate that Cs is potentially absorbed but that pH values have a strong effect on the activities of antioxidative enzymes in *A. undulatum.*

The results obtained for the MDA content in *A. undulatum* indicated the highest level of lipid peroxidation in plants treated with cesium acetate at pH 3 (Figure 2D), suggesting the negative effect of acid conditions rather than increased concentrations of cesium. However, no similar trend could be observed for plants in other experimental groups except that pH 6 combined with cesium acetate seemed not to affect the MDA content, i.e., lipid peroxidation in this case was relatively negligible compared to the control group. On the other hand, the lowest level of lipid peroxidation was documented in control plants at pH 8, while the addition of cesium acetate led to a slight increase of MDA. 

In control plant groups, changes in H_2_O_2_ followed a similar trend to MDA (Figure 2E). The lowest H_2_O_2_ content was measured in plants exposed to water supplementation at pH 8. Water supplementation with 0.5 mM cesium acetate resulted in a 49.4% lower H_2_O_2_ content at pH 4 compared to control plants. For the same cesium concentrations, pH 6 and pH 8 induced slightly higher H_2_O_2_ content than pH 4, while the highest H_2_O_2_ content was detected at pH 3. In general, 1 mM cesium acetate caused the highest increase in H_2_O_2_ for all tested pH values compared to control plants. On the other hand, plants exposed to 1.5 mM cesium acetate at pH 6 had similar values of H_2_O_2_ as control plants (Figure 2E). 

The lowest phenolic content was detected in control plants kept in distilled water at pH 6 (3.28 mmol GAE g^−1^) (Figure 2F). With an increase in cesium acetate concentrations (0.5, 1, and 1.5 mM), the TPC content also increased by 30.2%, 12.2%, and 61.6%, respectively. Changes in *A. undulatum*’s TPC content under other tested pH values combined with variation in cesium acetate concentrations were significantly different than for the control group. The TPC was the lowest at 1.5 mM cesium acetate in the case of pH 3, while at pH 4, the lowest value was observed at 1 mM cesium acetate (Figure 2F).

### 2.3. Effect of Short-Term Exposure for 24 h to Cesium Acetate in Experiment II (Heavy Rain Simulation)

The highest SOD activity was documented in the control plants at pH 6 when they were exposed to distilled water for 24 h (Figure 3A). Treatment with different concentrations of cesium acetate (0.5, 1, and 1.5 mM) at pH 6 decreased SOD activity by 37.8%, 56%, and 47.2%, respectively. Plants treated with 0.5 mM cesium acetate at pH 3 showed higher SOD activity, and, compared to the control group, it was increased by 38.3%. On the other hand, higher concentrations (1 and 1.5 mM) of cesium acetate decreased enzyme activity by 29.9% and 38.9%, respectively. Compared to the control group, the opposite was observed at pH 4, where the SOD activity at 0.5 mM decreased by 26.0%. The SOD activity at 1 mM cesium acetate was increased by 76.0% compared to the control plants, whereas at 1.5 mM cesium acetate, the SOD activity was the same as in the control group. No differences between SOD activity in *A. undulatum* kept for 24 h in water solution with 1 and 1.5 mM cesium acetate at pH 8 were observed (Figure 3A).

The 24 h exposure at pH 6 affected the increase of POX activity only at the highest tested concentration of cesium acetate (1.5 mM), and it was 56.8% higher than the value obtained in control plants (Figure 3B). On the contrary, the POX activity in *A. undulatum* kept for 24 h in water supplemented with cesium acetate at pH 8 decreased with increase of the cesium acetate concentration. The lowest activity was observed at the highest cesium concentration, and it was 17.9% lower than the value observed in the control plants. In general, at pH 3, the POX activity was higher in the presence of cesium acetate, except at 1 mM concentration, where the enzyme activity decreased by 5.4% compared to control plants. The POX activity measured in *A. undulatum* kept in a water solution supplemented with 0.5 mM cesium acetate at pH 4 decreased by 24.7% compared to the control group.

Similar to the results obtained for the light rain simulation experiment (Figure 2C), the CAT activity was the highest in plants treated with the solution at pH 6 (Figure 3C). However, with the increase of cesium acetate concentration (0.5, 1, and 1.5 mM), the CAT activity decreased by 36.8%, 58.7%, and 31.4%, respectively. The CAT activity follows a similar trend for the other tested pH values, but the obtained values are significantly lower than those obtained for pH 6 (Figure 3C). In general, the activities of antioxidative enzymes were affected more by pH values than the cesium present in the water. However, similar trends regarding enzyme activity were observed when plants were exposed to light and heavy rain, indicating that the duration of cesium and pH treatment did not drastically influence the enzymatic component of *A. undulatum*. 

The obtained results showed that the MDA content in *A. undulatum* plants kept in a water solution at pH 4 with cesium acetate supplementation did not change drastically (Figure 3D). The highest lipid peroxidation was documented in plants kept in the solution with pH 3, similar to the results obtained for light rain simulation (Figure 2D). However, after 24 h, the control plants at each pH value showed a similar MDA content, indicating that pH change had little effect on the lipid peroxidation when cesium was not applied (Figure 3D). On the other hand, at pH 6 and 8, cesium affected the decrease of MDA compared to the control group. The MDA content was reduced by 34.6%, 36.8%, and 29.8%, respectively, at pH 8 and by 5.1%, 34.9%, and 40.1% at pH 6, respectively, with increase in cesium acetate concentrations (Figure 3D).

Plants kept in a water solution for 24 h at pH 3 and pH 6 combined with 1 mM cesium acetate had the same H_2_O_2_ content (Figure 3E). The highest H_2_O_2_ content was documented in plants treated with 1.5 mM cesium acetate at pH 3 (9.4% higher compared to the control group). On the other hand, plants from the water solution supplemented with cesium acetate at pH 6 had the lowest H_2_O_2_ content (36% lower compared to the control group). The same was observed at pH 4 in combination with 1.5 mM cesium acetate, where the H_2_O_2_ content was 29.8% lower compared to the control group. The H_2_O_2_ content at pH 8 was the highest in control plants. In plants treated with cesium acetate (0.5, 1, and 1.5 mM), the content of H_2_O_2_ decreased by 37.1%, 32.3%, and 19.3%, respectively (Figure 3E).

After 24 h exposure to the water solution at pH 6, the highest TPC content was documented with the application of 1 mM cesium acetate (increased by 18% compared to control plants) (Figure 3F). Compared to the control group, a decrease in TPC was observed in plants treated with 0.5 and 1.5 mM cesium acetate by 11.2% and 21.5%, respectively. At pH 3, TPC decreased by 28.8%, 46.9%, and 17%, in all cesium-treated groups (0.5, 1, 1.5 mM), respectively. The highest observed TPC for the group kept in water at pH 8 was found in plants from a water solution supplemented with 1.5 mM cesium acetate. This TPC was increased by 5.1% compared to the control group. All plants kept in water solution at pH 4 and supplemented with cesium acetate showed higher TPC compared to the control group at the same pH. The highest TPC for plants kept for 24 h at pH 4 was in the case of 0.5 mM cesium acetate (1.85 mmol GAE g^−1^) (Figure 3F).

In general, the results obtained for light rain simulation (Figure 2) and heavy rain simulation (Figure 3) showed some comparable trends regarding antioxidative enzyme activities as well as MDA and TPC content. Such similarities indicate that the duration of cesium exposure was not the main factor leading to physiological changes in this research. However, the pH and cesium combined had some specific effects, but more experimental work is needed to come to clear conclusions for this species. 

In the case of short-term stress induced by cesium (experiment II), the pH value, and their interaction, it had a significant impact on the enzyme activity, lipid peroxidation, total phenolic content, and production of H_2_O_2_ (Table 1). 

## 3. Discussion

The uptake of cesium, as tested in this study, occurs both from the precipitates and from the substrate. The production of reactive oxygen species (ROS) and the changed activity of antioxidative enzymes were shown to be caused by the presence of cesium in the moss *Atrichum undulatum*. 

Elevated ROS concentration leads to changes in enzyme activities which remove ROS to establish redox homeostasis in the organism. Thus, SOD presents the first line of defense against ROS, converting the superoxide anion radical into H_2_O_2_, which is further converted into water and oxygen by CAT and POX [30]. Enzyme activities were reported to be related to the metal stress and overproduction of free radicals [31]. In the present model, moss *A. undulatum*, at pH 6, cesium exposure for 5 weeks leads to a decrease in SOD activity, indicating fewer free radicals over time. The SOD activity was several times higher at short exposure in rain simulation (experiment II), also decreasing with the time of cesium exposure at pH 6. This implies SOD to be the first level of response to short-term cesium exposure. In the research of Guo et al. [32], different patterns are shown to be present in two barley genotypes stressed with aluminum and cadmium at pH 6.5 and pH 4. In barley, either the same SOD activity was measured or the activity increased depending on the exposure duration and treatment type. After 24 h rain simulation, the CAT activity in control plants free of cesium at pH 6 increased by 60% compared to plants exposed for 2 h, suggesting the the postponed role of stress reaction of CAT reached peak activity after the SOD activity peak. As expected, the cesium uptake was lower from the medium even after longer exposure, as inferred from the CAT activity. Guo et al. [32] showed an increase in POX activity with an increase in exposure duration. In this research, the activity of POX showed a similar pattern to CAT but with a significantly lower magnitude. In contrast, Sun et al. [33] observed opposite trends of CAT and POX activities with the addition of lead and nickel nitrates to *Hypnum plumaeforme* Wilson. However, in this case, the experimental conditions were not fully controlled, and the moss was not free of cohabitant. In *A. undulatum*, the trends of POX and CAT activities after long-term exposure that lasted 5 weeks were opposite to *H. plumaeforme.* This could be a consequence of a diverse growth form, since *A. undulatum* is erect, and *H. plumaeforme* is a prostrate moss, suggesting slower transfer of cesium taken up from solid substrate cell to cell in acrocarpous moss, i.e., *A. undulatum.* In some other plant groups, like in stoneworts, namely in *Nitella pseudoflabellata* A. Brown, the activities of CAT and POX under cesium exposure were shown to participate in the activation of oxidative defense mechanisms. The concentration of H_2_O_2_ measured in *N. pseudoflabellata* under cesium exposure is in range with concentrations tested in this research [22]. The induction of antioxidant defense (SOD, POX, and CAT) under cesium treatment has also been shown in vascular halophyte *Sesuvium portulacastrum* L. [34] and *Brassica juncea* (L.) Coss [35]. The SOD activity in roots and shoots of halophyte *S. portulacastrum* is higher than the values measured in our experiments for long-term exposure, but this one was additionally stressed by salt. However, the SOD activity of *S. portulacastrum* under cesium treatment was not so different compared to the values obtained for short-term exposure. Further, the CAT activity in *A. undulatum* under conditions of long- and short-term exposure was several times higher than in roots and shoots of *S. portulacastrum* [34].

Increased production of O_2_·^−^ and H_2_O_2_ induces the disruption of cell membrane lipids known as lipid peroxidation. As a result of lipid peroxidation, malondialdehyde (MDA) is formed [36]. Experiments on *A. undulatum* showed that the presence of cesium does not lead to a drastic increase in lipid peroxidation and is unrelated to exposure time at pH 6. Lipid peroxidation as seen through measured MDA levels for 24 h rain simulation was documented to have an even lower level compared to 2 h rain simulation at pH 6. Measured MDA concentrations in all treatments were drastically lower than those measured under heavy metal stress with Cr, Cu, Zn, Ni, and Pb, both in bryophytes and tracheophytes [33,37]. The measured concentration of MDA in this study is more than ten times lower than the values measured in the roots and aboveground parts of *B. juncea* under cesium stress conditions [35]. Hydrogen peroxide (H_2_O_2_) is a non-radical ROS form that is synthesized in all cell compartments, including chloroplasts, peroxisomes, mitochondria, plasma membranes, and apoplast. It represents the most stable form of ROS. The toxicity of H_2_O_2_ in plant cells is reflected in its ability to form ·OH^−^ [38]. In moss *H. plumaeforme*, the dose-dependent increase of two ROS species, H_2_O_2_ and O_2_^−^, was observed [33]. The study of Choudhury and Panda [37] on the moss *Taxithelium nepalense* Broth revealed a similar trend of ROS accumulation under lead and chromium. The increase of H_2_O_2_ and O_2_^–^ in the moss cells was demonstrated to be dependent on the duration of the metal treatment [39]. On the other hand, levels of peroxide in the presence of cesium were very similar in the case of long- and short-term exposure at pH 6, except for the highest concentration, where the peroxide level was a bit higher in the case of long-term exposure. A similar H_2_O_2_ content was observed in *A. undulatum* in the case of short-term exposure at pH 3 for 2 h and 24 h. The lowest H_2_O_2_ activity was measured in plants kept for 24 h in heavy rain simulation treatments, i.e., supplemented with 1 mM cesium acetate. At pH 4, the H_2_O_2_ content for the highest cesium concentration after 24 h was more than two times lower compared to 2 h exposure. Meanwhile, at pH 8, the level of peroxide was very similar after 2 h and 24 h of exposure. The concentration of H_2_O_2_ measured in the aboveground parts of *B. juncea* under cesium supplementation [36] was higher than those measured herein in *A. undulatum*.

The cell anti-ROS defense system comprises both enzymes and non-enzymatic compounds. A high-stress environment increases the formation of ROS which activates defense mechanisms, e.g., SOD, CAT, POX, as well as phenolic compounds [40]. Plants in the control group grown on a solid medium (long-term exposure) produced more phenols than those under simulated 2 h light rain and 24 h heavy rain. Increasing cesium (at pH 6) leads to a decrease in TPC after 24 h, but their levels after 5 weeks are similar to those measured after 2 h of exposure. The exception was 1 mM applied cesium, where TPC did not increase after 5 weeks. Exposure to short-term stress (pH 3; 2 h to 24 h and increase of cesium acetate) decreased phenol production except at 1.5 mM cesium, where levels of phenol measured after 24 h were comparable to those obtained for the control group. An increase in stress exposure (both incubation time and cesium acetate concentration at pH 4) reduced the total phenolic content. Exposure of plants to pH 8 increased the production of phenols (as compared to control samples and exposure to 1.5 mM cesium acetate).

These findings imply that the plants at such concentrations lose the ability to cope with high amounts of cesium or some other additional mechanisms are activated. It further means that there is more than one level of stress response activated in the presence of cesium, and that next-level defense is activated after the previous one reaches its peak. However, this activation does not terminate the previous response completely—rather, they act synergistically. It can be inferred that the pleiotropic effect at higher cesium amounts is a possible response to cesium-caused stress in moss *A. undulatum*. 

The pH of the plant environment is an important factor controlling the process of biosorption of metals and radionuclides. It affects the surface charge of the biosorbent, the degree of ionization, and the speciation of the heavy metal in solution [41]. There are only a few publications on the role of pH in cesium availability, and they mostly refer to synthetic materials. According to Wang et al. [42], the concentration of species surface (uncharged surface groups as well as positively/negatively charged surface groups) can be obtained as a function of pH, influencing the cesium sorption. Possible mechanisms of the effect of pH on cesium sorption include increase of the variable negative charge, modification of metal speciation, displacement of the equilibrium of the surface complexion reaction, and competition of H_3_O^+^ for negative sites [43]. Saleh et al. [44] conducted research on the submerged vascular plant *Myriophyllum spicatum* L. showing that cesium sorption depends on the water pH value. Higher sorption of cesium was noted at pH 7 and pH 8. The same was noted for the plant *Azolla filiculoides* Lam. [41].

From the results obtained, we can infer that cesium affects the physiological traits of the moss *A. undulatum.* The pH did not affect the physiology of the moss, but it is assumed to change the antioxidative enzyme activities through protein conformation. The enzymes showed the highest activity in a slightly acidic environment (pH 6–7), and the changes at lower pH documented in this moss seem to be secondarily supported by the presence of cesium.

## 4. Materials and Methods

### 4.1. Plant Material and Culture Conditions

The plant material used in these experiments originated from an axenic culture of an in vitro-grown genotype of the moss *A. undulatum* from mountain Avala in Serbia [45,46]. Cultures of *A. undulatum* were grown on half-strength MS medium [47] enriched with sucrose (15.0 g L^−1^) to achieve optimal shoot production for further experiments. The plant material was maintained axenically in controlled laboratory conditions before and during experiments. The medium pH was adjusted to 5.8 before sterilization by autoclaving at 121 °C for 30 min. Ten-millimeter-long apical parts of the gametophores were used as experimental explants. Each explant represented a different clonal individual which was propagated in vitro from the same genotype; therefore, all plants shared the same genotype. Explants were planted vertically in such a way that one half (ca. 5 mm) was submerged in the medium and the other half outside of it. 

### 4.2. Experimental Design

The experiments conducted in this study aimed to simulate air pollution by cesium following different scenarios and to do so in controlled laboratory conditions free of xenic organisms. Two experiments were conducted, namely experiments I and II. In experiment I, long-term stress was simulated through cesium uptake from the solid substrate. In this type of experiment, individual plants were exposed to solid media containing three cesium acetate concentrations (0.5, 1, and 1.5 mM) for 5 weeks. In experiment II, short-term stress was simulated through cesium uptake from precipitates. In this case, individual plants were exposed to a water solution supplemented with different concentrations of cesium acetate (0.5, 1, and 1.5 mM) at different pH values (3, 4, 6, and 8) for 2 h and 24 h. Different durations of experiment II were set to simulate light (2 h exposure) and heavy rain (24 h exposure). Thus, the effects of the uptake from the substrate vs. rain simulation, different durations of exposure, varied cesium acetate concentrations (0.5, 1, and 1.5 mM), and different pH values (3, 4, 6, and 8) were investigated. The details of both experimental types are given below.

**Experiment I** (uptake from the substrate): The explants of young moss tips (10 mm) were grown for 5 weeks on a solid medium containing 0.5, 1, and 1.5 mM cesium acetate. The medium pH was adjusted to 6. Control plants were grown under the same conditions but without cesium. All plants were grown at 18 ± 2 °C and 60–70% humidity under cool-white fluorescent light with a 16 h photoperiod. 

**Experiment II** (rain simulation): (1) *Light rain simulation:* The explants of young moss tips (10 mm) were kept in water solution for 2 h at pH 3, pH 4, pH 6, and pH 8 with different cesium acetate concentrations (0.5, 1 or 1.5 mM) for each pH applied. After 2 h, plants were transferred to the solid basal medium without cesium and were grown, i.e., recovered for five weeks. Plants from the control group were kept for 2 h in distilled water with the pH adjusted to 3, 4, 6, and 8 and were then transferred to a solid basal medium free of cesium for five weeks. There were 4 control groups, one for each pH value. (2) *Heavy rain simulation*: The explants of young moss tips (10 mm) were kept in a water solution for 24 h at pH 3, 4, 6, and 8 with different cesium acetate concentrations (0.5, 1 or 1.5 mM) for each pH. After 24 h, plants were transferred to the solid basal medium without cesium and were grown for five weeks. Plants from the control group were kept in distilled water for 24 h with the pH adjusted to 3, 4, 6, and 8 and were then transferred to a solid basal medium free of cesium for five weeks. There were 4 control groups, one for each pH value. All plants were grown at 18 ± 2 °C and 60–70% humidity under cool-white fluorescent light with a 16 h photoperiod.

In the case of both experiments, there were 30 explants per each experimental group (3 petri dishes, 10 explants in each). Explants were randomly chosen from a pool of in vitro material subcultured at the same time, in order to eliminate the variability of different developmental stages. 

The biological effects of cesium were tested through the application of cesium acetate, which dissolves into cesium cation and acetate anion, the latter being shown to be harmless to plants and thus not masking the cesium effects, if any. The various pH values were tested in order to investigate whether the pH interferes with the cesium uptake in terms of the physiological traits of the moss tested.

### 4.3. Determination of Antioxidant Enzyme Activities

#### 4.3.1. Tissue Extract Preparation and Protein Content

Before the determination of protein concentrations and antioxidative enzyme activities, 1 g of frozen plant material was homogenized in liquid nitrogen. Crude proteins were extracted using a buffer containing 50 mM tris (hydroxymethyl)aminomethane (Tris), 1 mM ethylenediaminetetraacetic acid (EDTA), 30% glycerol, 1.5% polyvinyl polypyrrolidone (PVPP), 10 mM 1,4-dithiothreitol (DTT), and 1 mM phenylmethylsulfonyl fluoride (PMSF). The homogenates were centrifuged for 5 min at 12,000× *g* and 4 °C (Sorvall Heareus Biofuge Stratos Marshall Scientific). The protein content was determined from supernatants using the Bradford method [48]. Supernatants were stored at −70 °C until the enzymatic activity assays were performed. 

#### 4.3.2. Analytical Assays of Antioxidant Enzyme Activities

The activity of peroxidase (POX, U mg^−1^ of soluble protein) was assayed by measuring the increase in absorbance at 430 nm. The reaction mixture (3 mL) contained 50 mM potassium phosphate buffer (pH 6.5), 10 μL of enzymatic extract, 60 μL of 1 M pyrogallol as a hydrogen donor, and 30 μL of 1 M H_2_O_2_. Absorbance was measured using an Agilent 8453 UV-visible spectrophotometer. The activity of catalase (CAT, μmol H_2_O_2_ min^−1^ mg^−1^ of soluble protein) was determined spectrophotometrically by measuring the decrease in absorbance at 240 nm [49]. The reaction mixture (3 mL) contained 50 mM potassium sodium phosphate buffer (pH 7), 20 μL of enzymatic extract, and 30% H_2_O_2_. The activity of superoxide dismutase (SOD, U mg^−1^ protein) was determined according to Beyer and Fridovich [50]. The reaction mixture (3 mL) contained 100 mM potassium phosphate buffer (pH 7.8), 2 mM EDTA, 260 mM L-methionine, 1.5 mM nitroblue tetrazolium chloride (NBT), 0.04 mM riboflavin, and 0–50 μL of enzymatic extract. The mixture was kept under fluorescent light (Tesla Pančevo, 65 W) for 60 min at 25 °C. One unit of SOD was taken to be the amount of the enzyme where the NBT reduction (to blue formazan) ratio was 50%. The NBT reduction ratios were measured with a microplate reader (Multiscan Sky Thermo Scientific, Vantaa, Finland) at 540 nm. To determine enzyme activity, five measurements were performed for each of the treatments.

#### 4.3.3. Determination of Malondialdehyde and Hydrogen Peroxide Concentrations

Lipid peroxidation was quantified by measuring malondialdehyde (MDA) content in plant tissues following the method of Heath and Packer [51]. A total of 100 mg of fresh plant material from each sample was homogenized with 1 mL of 0.1% trichloroacetic acid (TCA). The homogenate was then centrifuged (15,000× *g* for 10 min), and the supernatant was added to 0.5 mL of 0.5% thiobarbituric acid (TBA) diluted in 20% TCA. The mixture was incubated at 95 °C for 30 min and rapidly cooled in an ice bath. The mixture was re-centrifuged, and the absorbance of the supernatant was recorded at both 532 nm and 600 nm absorbance using a spectrophotometer (Agilent 8453 UV/Vis Spectrophotometer, Agilent Technologies, Inc., Mulgrave, Australia). The malondialdehyde content was calculated using the following formula, and lipid peroxidation of plant samples is expressed as μmol MDA g^–1^ fresh weight:

MDA (μmol g^−1^) = [(A_532_ − A_600_)/155] × 1000 × Vol_extract_/Vol_aliquot_ × 1/FW(mg), where 155 is the molar extinction coefficient for MDA; Vol_extract_—volume of extract; Vol_aliquot_—volume of extract aliquot. 

For hydrogen peroxide (H_2_O_2_) estimation, 100 mg of fresh plant material was homogenized in TCA (0.1%) and centrifuged at 15,000× *g* for 15 min (Sorvall Heareus Biofuge Stratos Marshall Scientific). In the supernatant, 1 M potassium iodide and potassium phosphate buffer were used, as suggested by Velikova et al. [52]. The absorbance of the supernatant was recorded at 390 nm using a spectrophotometer. The H_2_O_2_ content was then calculated using the following formula: H_2_O_2_ (μmol g^−1^) = [(A_390_)/0.28] × 1000 × Vol_extract_/Vol_aliquot_ × 1/FW (mg), where 0.28 is the molar extinction coefficient for H_2_O_2_; Vol_extract_—volume of extract; Vol_aliquot_—volume of extract aliquot. For the determination of MDA and H_2_O_2_ concentrations, five measurements were performed for each of the treatments.

#### 4.3.4. Spectrophotometric Analysis of Total Phenolic Content (Folin–Ciocalteu Test)

Total phenolic content (TPC) was determined using the Folin–Ciocalteu test (FC test) [53]. Total phenolics in plant extracts react with the Folin–Ciocalteu reagent, forming a blue-colored complex that can be spectrophotometrically quantified. For this purpose, 200 mg of plant material was homogenized in liquid nitrogen with 2 mL of 96% ethanol. The extract was incubated for 60 min at room temperature and then centrifuged at 12,000× *g* for 15 min. The supernatant was transferred to new Eppendorf tubes. The reaction mixture contained 300 μL of FC reagents solution (diluted with water in a ratio 2:1 *v*/*v*), 1340 μL of deionized H_2_O, and 60 μL of supernatant. The mixture was quickly vortexed and left at room temperature for 5 min. Then, 20% Na_2_CO_3_ was added into the mixture and left at room temperature for 90 min in dark conditions. Absorbance was measured at 765 nm using a plate reader. Gallic acid was used as a standard phenol. The total phenolic content was expressed as mmol of gallic acid equivalents/g extract. For the determination of TPC, six measurements were performed for each of the treatments.

### 4.4. Statistical Analysis

Complete statistical analysis was conducted using the R programming language (v. 4.3.1) [54]. The significance level (α) was set at 0.05 for all experiments. Preliminary exploration of the data consisted of using the Shapiro–Wilk normality test and Lavene’s test of homogeneity of variance, which revealed that not all experimental groups followed a normal distribution and that homogeneity of variance was violated across the groups. Thus, non-parametric statistics were applied. In the long-term exposure experiment, the Kruskal–Walls test was used for comparison of the groups, after which Dunn’s multiple comparisons test with the Benjamini–Hochberg *p*-value adjustment method was applied. Regarding the short-term exposure experiments, the aligned rank transform (ART) procedure was applied to implement the nonparametric factorial ANOVA [55,56]. ART was implemented for each parameter and exposure duration using the ARTool R package [57]. Factorial models were created using the art() function, after which the significance of these effects was evaluated using the anova() function. The contrast tests were conducted using the art.con() function of the ARTool package.

## 5. Conclusions

Our study showed that the presence of cesium leads to the formation of ROS, which was demonstrated by changes in antioxidative enzyme activity and different levels of TPC. The applied concentration of cesium affected the physiological traits of the moss *A. undulatum.* At pH 6, SOD showed high activity in the case of short-term cesium exposure, while its activity was lower for long-term cesium exposure. If we observe the exposure time, POX and CAT had opposite activity trends compared to SOD at pH 6. This confirms SOD as the first line of defense against ROS, while POX and CAT are activated later. We assume that changes in enzyme activity at other pH values are related to enzyme conformation that is disrupted at pH other than optimal (suboptimal or pH that disrupts enzyme structure leading to its inactivation or degradation). Understanding the physiological disturbances in mosses after cesium uptake will help in comprehension of the magnitude of the harmful effects of cesium produced in the plant body.

## Figures and Tables

**Figure 1 plants-13-00054-f001:**
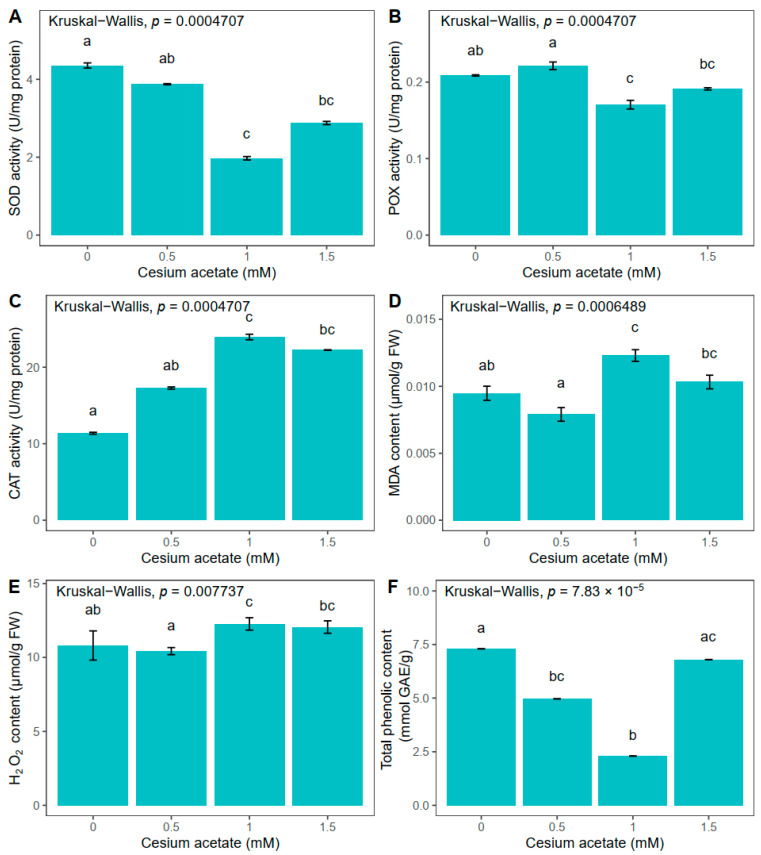
Oxidative stress-related parameters in *Atrichum undulatum* exposed for 5 weeks to different concentrations of cesium acetate at pH 6 (experiment I): superoxide dismutase (SOD) activity (**A**), peroxidase (POX) activity (**B**), catalase (CAT) activity (**C**), malondialdehyde (MDA) content (**D**), hydrogen peroxide (H_2_O_2_) content (**E**), and total phenolic content (**F**). Data are represented as the mean ± standard deviation. Different letters above the bars represent statistically significant differences (*p* < 0.05) between the experimental groups.

**Figure 2 plants-13-00054-f002:**
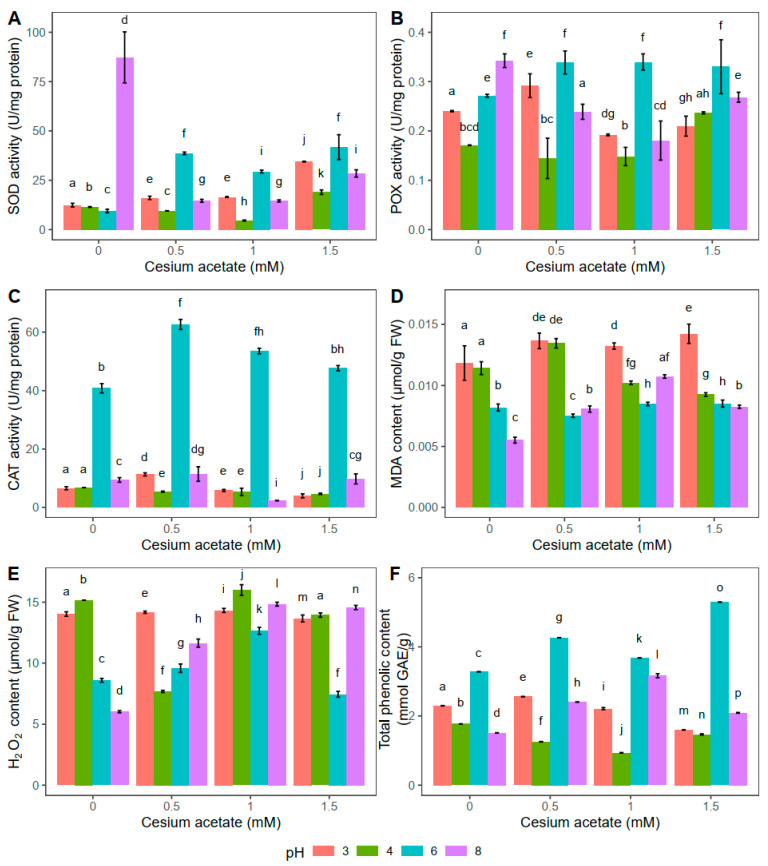
Oxidative stress-related parameters in *Atrichum undulatum* exposed for 2 h to different concentrations of cesium acetate at different pH (3, 4, 6, 8) (experiment II—light rain simulation): superoxide dismutase (SOD) activity (**A**), peroxidase (POX) activity (**B**), catalase (CAT) activity (**C**), malondialdehyde (MDA) content (**D**), hydrogen peroxide (H_2_O_2_) content (**E**), and total phenolic content (**F**). Data are represented as the mean ± standard deviation. Different letters above the bars represent statistically significant differences (*p* < 0.05) between the experimental groups.

**Figure 3 plants-13-00054-f003:**
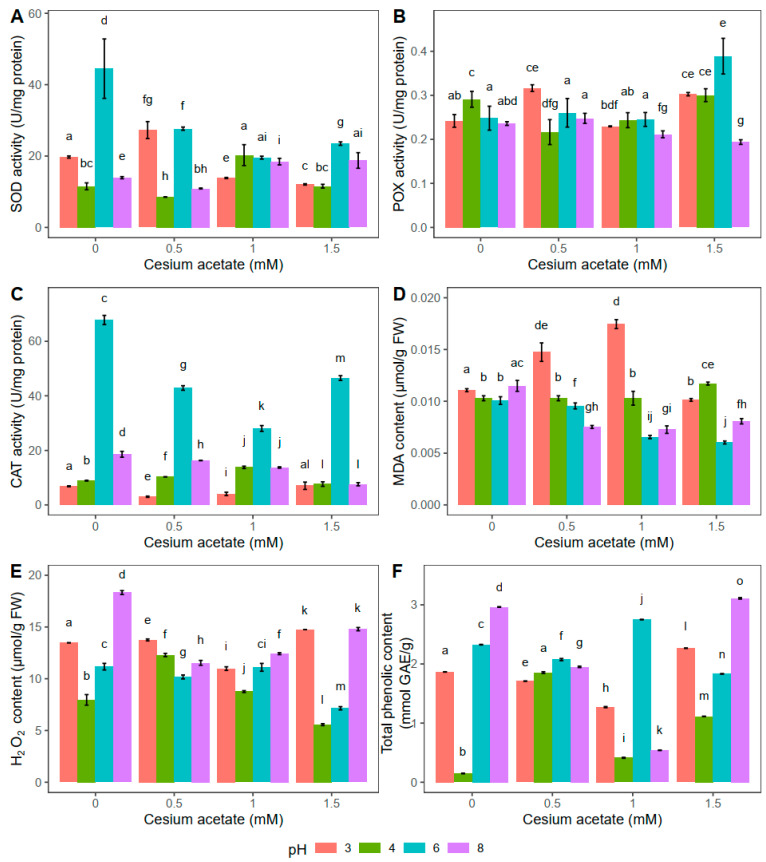
Oxidative stress-related parameters in *Atrichum undulatum* exposed for 24 h to different concentrations of cesium acetate at different pH (3, 4, 6, 8) (experiment II—heavy rain simulation): superoxide dismutase (SOD) activity (**A**), peroxidase (POX) activity (**B**), catalase (CAT) activity (**C**), malondialdehyde (MDA) content (**D**), hydrogen peroxide (H_2_O_2_) content (**E**), and total phenolic content (**F**). Data are represented as the mean ± standard deviation. Different letters above the bars represent statistically significant differences (*p* < 0.05) between the experimental groups.

**Table 1 plants-13-00054-t001:** Factorial ANOVA results for the main effects of cesium concentration (C), pH value (pH), and interaction effect (C × pH) on the oxidative stress-related parameters in *A. undulatum* plants exposed to stressors for 2 h and 24 h. CAT—catalase activity; POX—peroxidase activity; SOD—superoxide dismutase activity; TPC—total phenolic content; MDA—malondialdehyde content; H_2_O_2_—hydrogen peroxide content.

	CAT		POX		SOD		TPC		MDA		H_2_O_2_	
Variable	2 h	24 h	2 h	24 h	2 h	24 h	2 h	24 h	2 h	24 h	2 h	24 h
C	39.56 *	156.46 *	14.92 *	37.99 *	80.76 *	36.29 *	168.76 *	401.48 *	40.11 *	35.78 *	155.62 *	198.15 *
pH	114.27 *	323.23 *	138.79 *	34.61 *	162.21 *	159.03 *	402.27 *	401.44 *	117.01 *	209.62 *	324.74 *	334.81 *
C × pH	88.44 *	223.95 *	21.49 *	32.53 *	117.60 *	224.35 *	1296.86 *	1093.28 *	51.62 *	172.16 *	159.21 *	437.98 *

Values represent the F values, with the asterisks denoting the corresponding level of statistical significance * *p* < 0.001.

## Data Availability

Data are contained within the article.

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
