# Peer review of "Effects of Cesium on Physiological Traits of the Catherine’s Moss *Atrichum undulatum* Hedw."

_plants, 2023, doi:10.3390/plants13010054_

Round 1

Reviewer 1 Report

Comments and Suggestions for Authors

#1 General comment:

The effects of caesium on physiological traits of the Catherine's moss Atrichum undulatum Hedw were studied in this study.

The response to the added amount of caesium acetate was inconsistent, suggesting the presence of factors that were not accounted for. And needs further explanation as mentioned by the authors.

#2 line 84: Cs-135 is also detectable. Cs-134 is not a fission product but an activated product.

#3 line 91: "Several MeV" is ambiguous. For internal exposure, the biological effect per unit energy delivered may be greater at lower energies. In any case, it may not be relevant to the discussion here on this matter since it has little to do with the essence of this paper.

#4 Figure 1 and 2: The results of Dunn's multiple comparison test should be presented in pairs that are significant. Could the pairs here be the amounts of different additions of cesium acetate? What do [a] and [ab] indicate here as "the experimental groups"?

#5 Figure 1 and 2:  An explanation of what the error bar indicates is needed.

#6  Table 1: The horizontal line is broken in the middle.

#7  The Kruskal-Wallis test was used to compare groups, but can't the amount of cesium acetate added be treated as a quantity rather than a categorical variable, even though the effect might be greatest at 1 mM?

#8  line 139: “Moreover, 1 mM of cesium acetate led to the greatest changes in the enzyme activities as well as in the TPC suggesting that plants engage antioxidative mechanisms to reduce the oxidative stress that potentially arose from the Cs presence in the solid media.”

Is it reasonable that 1.5mM has a weaker effect?

#9  line 220: “2.2” is duplicated. It would be “2.3” or use sub section.

Reviewer 2 Report

Comments and Suggestions for Authors

This paper seems to be the first to describe actual effects of Cs on bryophyte physiology, comparing results at 4 pH levels and effects of substrate vs precipitation/deposition effects on 6 physiological parameters.  The paper is well written and I only found one unclear statement.  The methods seem to be appropriate except that the following information is lacking:

The methods of selecting explants and replication are not explained.

How many field populations are represented?

Did each explant represent a different plant?  Or did the cultures represent just one genotype?

How many cultures were there?

How were replicates chosen from among the explants?

How many individual explants were present in one replicate?

Were the treatments replicated, or were all replicates for a treatment in the same container?

How were the explants randomized?

The introduction mentions potassium and potential competition, so I am a little disappointed that levels of potassium were not compared and implications for effects on potassium were not discussed.

I was also disappointed that there was no mention of color changes, even if none, or morphological changes.  Perhaps exposure was too short?

I did not check the references, but there seem to be some style errors.  Some dates are not in bold.

Comments on the Quality of English Language

Well written.
